# The Impact of Cellular Proliferation on the HIV-1 Reservoir

**DOI:** 10.3390/v12020127

**Published:** 2020-01-21

**Authors:** Maria C. Virgilio, Kathleen L. Collins

**Affiliations:** 1Cellular and Molecular Biology Program, University of Michigan, Ann Arbor, MI 48109, USA; mcvirgil@umich.edu; 2Department of Internal Medicine, University of Michigan, Ann Arbor, MI 48109, USA; 3Department of Microbiology and Immunology, University of Michigan, Ann Arbor, MI 48109, USA

**Keywords:** HIV-1, latency, HSPC, T cell, proliferation, differentiation, integration, provirus

## Abstract

Human immunodeficiency virus (HIV) is a chronic infection that destroys the immune system in infected individuals. Although antiretroviral therapy is effective at preventing infection of new cells, it is not curative. The inability to clear infection is due to the presence of a rare, but long-lasting latent cellular reservoir. These cells harboring silent integrated proviral genomes have the potential to become activated at any moment, making therapy necessary for life. Latently-infected cells can also proliferate and expand the viral reservoir through several methods including homeostatic proliferation and differentiation. The chromosomal location of HIV proviruses within cells influences the survival and proliferative potential of host cells. Proliferating, latently-infected cells can harbor proviruses that are both replication-competent and defective. Replication-competent proviral genomes contribute to viral rebound in an infected individual. The majority of available techniques can only assess the integration site or the proviral genome, but not both, preventing reliable evaluation of HIV reservoirs.

## 1. Introduction

Human immunodeficiency virus (HIV) is a chronic viral infection that causes acquired immunodeficiency syndrome (AIDS) if left untreated. HIV primarily targets the host immune system by infecting CD4+ T cells, which often leads to the death of the infected cell. With the development of antiretroviral therapy (ART) to block new infections, it was hoped that with time, all of the HIV infected T cells would be eradicated either through immune mediated clearance or cytotoxic effects from the replicating virus. However, a discordance between the amount of HIV mRNA and proviral (genomic, integrated DNA) sequences was discovered in that there was much more HIV DNA than mRNA. This indicated HIV could survive in an integrated but latent form. The main cellular reservoir was discovered to be resting memory CD4+ T cells [1,2], although other reservoirs have since been discovered. Soon after, researchers calculated the mean half-life of latently-infected resting memory T cells to be nearly 44 months and that clearance of the latent reservoir could take 70 years or more [3,4].

HIV relies upon virally encoded reverse transcriptase (RT), an enzyme that is capable of reverse transcribing the viral genomic mRNA into a double-stranded DNA genome that is transported into the cell nucleus and permanently integrated into the host chromosome. As RT is particularly error-prone, causing approximately one error per 10^4^ nucleotides (reviewed [5]) it is unlikely that any two ~10,000 nucleotide viral genomes will be exactly identical after reverse transcription. Sequencing of latent proviral sequences revealed many cells with the same genetic sequence in multiple cells prompting an investigation into how this could be. Because it is also extremely unlikely that any two HIV integration sites are exactly the same, integration site analysis and sequencing of HIV DNA genomes have been used to identify cases in which cellular proliferation is responsible for the production of exact copies of HIV in multiple cells. Each of these methods has their strengths and limitations. The large majority of proviral sequences in an individual are defective [6], meaning they are incapable of producing virus that can infect new cells. Both defective and replication-competent HIV sequences have been found in clonally expanded cells [7,8,9,10,11,12]. In addition to CD4+ T cells, HIV targets multiple CD4+ cell types in the hematopoietic compartment including macrophages and hematopoietic stem cells, each with a different proliferative potential and ability to contribute to maintenance of the latent reservoir [11,12,13,14,15,16]. 

## 2. Cell Type, Proliferation Potential, and Growth Stimulus

### 2.1. Introduction to HIV Infection in Patients

The potential for HIV latency was discovered several decades ago in patients receiving ART. First predictions of time to cure with continued therapy were approximately 2–3 years [17] based on initial results showing a rapid decrease in plasma virus after initiation of ART [18] and a short half-life of actively infected CD4+ T cells [17]. These initial estimates proved wrong due to the presence of cells with a much longer half-life. Further in vitro studies showing that non-virus producing T cells could be stimulated in culture to produce virus provided definitive evidence for the existence of a longer-lived latent reservoir. The major reservoir was thought to be long-lived memory T cells [2,3,17,19,20]. There is now evidence that other long-lived cells including naïve T cells, stem cell memory T cells [15,21] and hematopoietic stem and progenitor cells (HSPCs), contribute to the reservoir [16,22]. Virus may rebound upon interruption of ART through cellular stimulation and differentiation from cells within the latent reservoir. However, T cells and HSPCs have very different proliferative potentials and differentiation capabilities that contribute to viral latency and maintenance of the viral reservoir. The extent to which other long-lived cells such as tissue resident macrophages contribute to the reservoir is unknown. More work is needed to fully characterize the impact of noncanonical cellular subsets to maintaining the reservoir.

In the bloodstream, HIV targets cells of interest through recognition of its primary cellular receptor, CD4, in addition to one of two co-receptors: CXCR4 and CCR5 by the HIV encoded envelope (Env) protein. It has been suggested that all transmitted founder viruses—the initial viruses that establish infection—are CCR5-tropic (reviewed [23,24]), but eventually CXCR4-tropic (X4) viruses predominate in the most infected individuals [25,26,27,28]. Exceptions to CCR5-tropic (R5) founder viruses are well established and often found in CCR5−/− individuals [29,30]. Homozygous loss of CCR5 is usually due to a naturally occurring 32 base pair deletion in CCR5 (CCR5Δ32), and homozygous individuals are highly resistant to HIV infection [30,31,32]. Both X4-tropic and dual-tropic viruses (with Env that can recognize both CCR5 and CXCR4) have been found in CCR5Δ32 homozygous individuals [33,34]. Although CXCR4 is more widely expressed in the hematopoietic compartment, both CCR5 and CXCR4 are expressed on CD4+ T cells and HSPCs [12,16,35,36]. CCR5 is highly expressed on macrophages and activated T cells, but very little is found on quiescent T cells [37,38,39,40]. For HSPCs, the more primitive stem cells express less CCR5 than progenitors, however proviral sequences that express X4 and R5-tropic Env have been identified in both stem cell and progenitor populations [12,16,35].

### 2.2. Hematopoietic Potential

HSPCs possess enormous proliferative potential, are extremely long lived, and differentiate into all the hematopoietic lineages through hematopoiesis. Hematopoiesis is the process of building and maintaining the entire hematopoietic compartment through many stages of differentiation starting from the most primitive hematopoietic stem cells (HSCs). Differentiation initially proceeds from the long- and short-term HSCs, which divide and differentiate into progenitors that eventually terminally differentiate into all the cells that make up the hematopoietic system, including dendritic cells, macrophages, erythrocytes, megakaryocytes, B and T cells. With each step through the differentiation process, stemness is lost along with proliferative potential (reviewed [41,42]). Although most differentiated cells in the hematopoietic compartment are tissue-resident, HSPCs remain primarily quiescent in the bone marrow, except during times of stress or need (reviewed [43]).

HSPCs express CD4 at low levels, are capable of being infected with both X4 and R5-tropic HIV in vitro, and proviral sequences have been identified in HSPCs from patients [11,12,16,22]. Identical proviral sequences, including the same insertion site in the patient genome, were found in multiple compartments of the hematopoietic lineage including HSPCs, PBMCs (peripheral blood mononuclear cells), and BMMCs (bone marrow mononuclear cells). A subset of these clonal sequences was matched to expressed cell-free virus [11]. Some cells with clonal proviral sequences matching provirus from HSPCs were not CD4+. The virus detected was defective such that natural infection could not have occurred. These infected lineages could only have arisen through differentiation into multiple lineages from a single, more primitive cell [12]. This demonstrates the potential of a single infected HSPC to be infected, proliferate, and differentiate into several lineages while also carrying provirus (Figure 1a).

HSPCs are rare and HIV infected HSPCs are even rarer (approximately 2.5 proviruses per million HSPCs) with less than 1 in a million HSPCs expected to carry a replication-competent provirus [12]. Latency is preserved in the most quiescent HSPCs and differentiation is associated with higher rates of active infection [44]. A single HSPC harboring an intact virus has great potential to spread provirus through daughter cells and expressed virus to form new infections (Figure 1a). The high proliferative potential, long lifespan, and capability of populating the body with infected daughter cells of all lineages, not just T cells, makes HSPCs a unique reservoir with exceptional potential for maintaining the viral reservoir.

### 2.3. T Cell Reservoirs

Once T cells have committed to the CD4 lineage, the naïve T cells (T_N_) have the greatest proliferative capacity of the CD4+ T cells and lose their proliferative potential as they differentiate into the polarized, functional subsets: T_H1_, T_H2_, T_H9_, T_H17_, T_FH_, T_reg_ (reviewed [45]). After the peak of infection and clearance of foreign antigen, greater than 90% of the T_H_ population dies. The remaining cells convert into long-lived memory cells that are primarily quiescent and do not rely on peptide-bound major histocompatibility complex II stimulation for maintenance (reviewed [46]). T cells appear to be most susceptible to HIV infection after activation but before initiation of quiescence [1], while the latent reservoir in T cells is most likely established during the effector to memory transition [47]. These memory cells then rely upon homeostatic proliferation for maintenance until they are re-activated (Figure 1b).

Within the memory compartment, there are a number of populations of CD4+ T cells with differing proliferative and effector capabilities that have been shown to play a role in maintaining the viral reservoir. The long-lived memory compartment generally follows a sequential differentiation program from T_N_ < T_SCM_ < T_CM_ < T_TM_ < T_EM_ < T_TE_ where proliferative potential and life-span are lost as differentiation state is gained (Figure 2) [48,49]. Among T cell subsets, T_N_ have the greatest proliferative potential but the slowest rate of growth with an estimated proliferation rate of only 0.2% per day [50,51]. Of the two major memory subsets, effector memory T cells, T_EM_, have rapid proliferation rates of 4.7% per day compared to 1.5% for T_CM_ [50]. T_EM_s are short lived cell populations needing continuous cell replenishment in vivo [50] and low proliferative potential compared to T_CM_ [52,53]. To grow and maintain their existence, CD4+ T cells rely on IL-7 and IL-15 for homeostatic proliferation [52,54,55]. Experiments stimulating proliferation of latently-infected T cells using IL-7 surprisingly did not result in viral reactivation alone [56,57] or together with IL-2 [58] (IL-2 reviewed here [59]), indicating that homeostatic proliferation can maintain the reservoir over time. However, a combination of IL-15 and the pan-histone deacetylase inhibitor, vorinostat, [60] or antigenic stimulation using anti-CD3/anti-CD28 can efficiently reactivate latent HIV in cultured central memory T cells and lead to virus-induced cell death [57,58]. Therefore, cellular stimulation signals and epigenetic drugs are capable of reactivating the virus in vitro and may be able to lessen the viral reservoir in patients, however, clinical trials using vorinostat have thus far not succeeded in decreasing the size of the latent viral reservoir [61].

Central memory T cells, T_CM_, are classically considered the primary reservoir of latent HIV infection. However, clonally expanded latent infections have been found in many memory T cell subsets [7,9,15,21,62]. CD4+ T memory stem cells, T_SCM_, are the most recent addition to the T cell memory compartment and are derived from naïve T cells. They are capable of maintaining a stem-like phenotype in addition to having central memory attributes [63,64]. Recent work has demonstrated T_SCM_ are more susceptible to in vitro infection using a GFP-encoding vesicular stomatis virus G protein (VSV-G) pseudotyped HIV and similarly susceptible with a GFP-encoding R5-tropic HIV isolate compared to T_CM_ even though T_SCM_ have slightly less CCR5 cell surface expression [21]. While T_SCM_ represent a much smaller fraction of the memory compartment they are proportionately more likely to contain HIV DNA than T_CM_ or T_EM_ [21]. Identical provirus sequences found in T_SCM_ were also in more differentiated T cell subsets including T_CM_ and T_EM_, demonstrating T_SCM_ contribute to the latent reservoir through both proliferation and differentiation, though the integration sites were not determined to confirm clonality [21]. When considering the ability of proviruses found in memory T cell subsets to produce virus either by viral outgrowth assays or by sequencing of near full length (NFL) viral genomes T_SCM_ [21], T_EM_, and T_N_ [15] had more replication-competent provirus sequences than T_CM_, making their effective contribution to the functional HIV reservoir much larger [15,21]. Their findings are surprising considering central memory T cells harbor more total HIV DNA. T_SCM_ and T_N_ both have greater proliferative potential than T_CM_, but naïve T cells have largely been dismissed as contributing to the latent reservoir because they are difficult to infect in vitro, are primarily quiescent, and express low levels of CCR5 before activation [37,38]. It is possible that a higher proportion of functional genomes are found in quiescent T cell subsets because viral latency is tighter and protects the cells from cytopathic effects of activated functional proviral genomes. Interestingly, HSPCs, which are also quiescent in vivo, similarly harbor a higher proportion of functional proviral genomes compared to more differentiated cells [11].

## 3. HIV Sequence Integrity

Many HIV sequences are clonal, and the large majority of proviruses are defective [6,65]. Although some work has shown only cells carrying defective proviruses can clonally expand [7], cells with replication-competent provirus can be clonal with potential to maintain spreading infection and rebound with interruption of therapy [8,11,14,66,67].

A replication-competent provirus can be thought of as coding for all the necessary structural proteins – gag, pol, and env, and genomic features necessary for transcription including cis regulatory sequences in the 5’UTR such as the major splice donor site (MSD) used in all the spliced viral RNAs, promoter, and transcriptional start site (reviewed [68]). Ultimately, replication-competence is measured by the ability of a provirus to code for new virions that are capable of infecting new cells. Replication-incompetent, or defective proviral genomes can take many forms. They generally have acquired large deletions impacting necessary genomic features including the major structural proteins and regulatory regions. They can also be classified as hypermutated where proviral genomes contain a multitude of point mutations, likely as a result of APOBEC3G activity (reviewed by [69]).

### 3.1. Clonality and HIV Sequence Integrity in T Cell Subsets and HSPCs

As summarized above, CD4+ T cells have many levels of differentiation. The contribution of each differentiated subset to the functional latent reservoir has only recently begun to be more comprehensively resolved. Lee et al. performed massive single-genome, near-full-length next-generation sequencing on HIV DNA from bulk memory and functionally polarized memory CD4+ T cells: T_H1_, T_H2_, T_H9_, T_H17_, and T_Hneg_ (T_H_ cells that did not fit into the other T_H_ subsets) [14]. Of the clonally expanded cells with intact provirus in the polarized T cell subsets, the majority (11%) of the intact sequences were found in the T_H1_ subset. Only defective clonal sequences were found in T_H9_ cells. Additionally, 4% of sequences in memory CD4+ T cells were genome intact. The intact clonal sequences were phylogenetically mixed, suggesting the sequences did not originate from a dominant plasma strain (active infection), but from proliferation of single cells that become clonal. Viral outgrowth assays confirmed the clonal sequences classified as fully intact were replication-competent [14]. Similarly, Hiener et al. found approximately 5% of recovered sequences from six individuals were replication-competent within CD4+ T cell memory subsets [15]. More intact viral genomes were found in T_EM_, T_TM_, and T_N_ than T_CM_, even though HIV DNA was more prevalent in T_CM_. Interestingly, the authors identified a clone in one of their participants where 32% of all the sequences from T_EM_/T_TM_ cells were identical. The coding region was fully intact except for a deletion in the packaging loop and MSD site [15].

Compared to T cells, HSPC-associated HIV genomes appear more likely to be functional. Based on near-full-length genome amplification and sequencing, approximately 28% of proviral genomes from HSPCs were estimated to be replication-competent [11] compared to 2%–12% for CD4+ T cells [6,14,15,65]. Remarkably, HSPC-derived near-full-length genomes often matched free virus in the bloodstream (60%), otherwise known as peripheral virus (PV). This ‘‘in vivo outgrowth assay’’ provides strong evidence that HSPCs can harbor infectious HIV. The functionality of one of the near-full-length HSPC-associated proviruses was further confirmed by demonstrating virus produced from the reconstructed genome could spread in the CCR5 expressing T cell line, MOLT4/CCR5 [11]. Based on these studies, HSPCs, which have the greatest proliferation potential of all the latent HIV reservoirs (Figure 1a; Figure 2), harbor intact proviruses, and contribute significantly to residual PV.

Clonally amplified HIV proviral genomes with signature deletions have appeared in many reports. In particular, a number of groups have reported clonal amplification of proviral genomes containing large deletions removing the MSD site or other regulatory sequences in the 5’ region [11,15]. Pinzone et al. aligned whole genome sequences collected from four individuals to look for common genomic changes in donors over time [70]. They classified defective proviruses based on preservation of the two strong splice donor sites in HIV. The first strong splice donor (SD1) is just upstream of *gag* and is part of the canonical splice pathway, used for all proteins except Gag/Pol. The second, splice donor 4 (SD4), is further downstream and falls just before *vpu* [71,72,73,74]. Sequences with both splice donor sites intact decreased the most over time, followed by those with SD1, and finally SD4. Sequences with neither did not change at all. Unsurprisingly then, sequences preferentially expressing Gag/Pol (those lacking SD1) became more abundant over time, suggesting these deleted proviruses experience less negative selection pressure [11,70]. Their last finding was the most surprising of all. They demonstrated HIV could splice to downstream host genes including *STAT5B* [75], and in vitro experiments showed a preference for HIV-host chimeric transcripts spliced from SD4 [70].

### 3.2. Using HIV Sequences to Determine Clonality and Decay

Our understanding of the latent reservoir has changed significantly since it was first reported, as has our understanding of clonal expansion and proliferation of provirus-containing cells. Proliferation is not explained by acquisition of drug resistance mutations in *pol* for participants on ART [10]. In HIV infected individuals on ART, the majority of proviral sequences are clonal due to proliferation, and the frequency of clonally expanded cells increases with time as non-clonal cells decrease (Figure 3) [7,9,10,76]. For example, Simonetti et al. showed a replication-competent provirus from a clonal CD4+ T cell represented 13% of all CD4+ T cells in an infected individual. The site of integration is in a region of the human genome that is mapped to two different places in the reference. In addition, the replication-competency and spreading ability were confirmed by PCR-amplifying the viral sequence as two partially overlapping genomic products and co-transfecting the fragments into cells, then using the produced virus to infect CD4+ T cells from HIV negative donors in vitro [8].

Estimates of total HIV DNA and replication-competent sequences in an HIV-infected person are upwards of 10 million cells distributed amongst hundreds to thousands of clones [9,77]. Reeves et al. broke down the clonal population into estimates for large and small clone contributions [9]. The 100 largest clones in all their participants had approximately 10^5^ associated cells. Large clones with replication-competent provirus had an average of 10^4^ cells each. The average infected individual also had approximately 10^7^ total DNA-containing smaller clones with fewer than 1000 associated cells, and 10^4^ small clones with replication-competent provirus [9]. Even though the total amount of HIV DNA does not decrease over time, intact proviruses do. Intact proviruses decayed at a rate of −0.38 to −0.2/year with a half-life of 1.8 to 3.4 years in individuals on ART for >10 years [70]. Additionally, some recent work proposes sex-specific differences in the size and composition of the latent viral reservoir [78,79] (sex-specific infection differences reviewed [80]). Despite the gradual loss of intact proviruses over time in treated people, the quantity of cellular clones with intact, replication-competent provirus, and sometimes extreme proliferation rates, challenges previous perceptions of the contribution of proliferating clones containing non-replicating HIV DNA. Instead, future cure strategies must focus on cells with the ability to maintain themselves by proliferation if they will be successful at clearing HIV from the body.

## 4. Methods of Viral Spread

### 4.1. Cell-to-Cell Viral Spread

Infection is initiated by either a small number of virions or a single virion (reviewed [81]). Once infection is established, there are two main methods of viral transfer: cell free and cell-to-cell transmission (reviewed [82]). Cell free virions circulate in the bloodstream and can rapidly disseminate. Cell-to-cell transmission involves contact between at least one infected cell and another uninfected cell. It is a much slower process, but the overwhelming majority of new infections are from cell-to-cell spread and not cell-free virus [83,84]. As an antigen presenting cell, infected macrophages efficiently transfer HIV to nearby T cells through direct interaction [85,86,87]. Recently, a host factor involved in cell aggregation, activated leukocyte cell adhesion molecule (ALCAM), was found to mediate cell-to-cell transmission of HIV to T cells [88]. HIV proteins can also contribute to cell-mediated transfer. HIV Env is expressed and trafficked to the cell surface during active infection where it can interact with HIV receptors on uninfected cells, sometimes fusing cells together into syncytia [89]. Cell-to-cell spread minimizes the exposure of virus to the host immune system and lessens selective pressures [90].

### 4.2. Viral Evolution and Insufficient ART

Viral evolution in tissues with sub-optimal ART concentrations has been suggested as a potential source of residual viremia, continued viral evolution, and an important means of maintaining the viral reservoir in infected individuals [91]. However, the overwhelming majority of available evidence strongly suggests this is unlikely [13,92,93,94,95,96,97,98,99]. Instead, viral evolution early after initiation of ART may be an artifact of sampling long-lived cells that were generated before the initiation of ART or during acute infection [9,98,99,100,101]. After one week of ART, clonal expansion is primarily responsible for driving viral persistence and not viral evolution [9,13,96]. Single-genome sequencing (SGS) (Table 2) of samples taken from patients who initiated ART shortly after becoming infected shows there is little change in the provirus sequences after several years on ART, further providing evidence that tissues, or “sanctuary sites,” with lower drug concentrations do not contribute to maintenance of the viral reservoir [77].

If suboptimally treated tissues exist and contribute to persistent viremia, then increasing the dosage or adding new treatments to current regimens should limit viral evolution by reaching any untreated or under-treated tissues. However, studies testing the addition or increase in dosage of Raltegravir, an integrase inhibitor, to ART regimens have disagreed on whether modifying ART is helpful. One study claimed Raltegravir did not alter residual viremia in the gut [102] or alter viral evolution [97] while another group found Raltegravir added to normal ART regimes allowed the detection of unintegrated HIV DNA that could only be present in the setting of active viral replication [103]. Nevertheless, there is little evidence that any low level replication that may occur, significantly changes viral production or promotes viral sequence alterations towards drug-resistant variants [93,102]. Viral evolution is unlikely with standard ART regimes.

### 4.3. Expression from Unintegrated Virus

In addition to the stably integrated form, HIV proviral DNA can exist in labile form—that is, as an unintegrated complex—and some have speculated that labile forms contribute to persistent viremia. However, these forms of HIV are extremely unstable and only last up to a couple of days [104]. They can, however, last longer in quiescent cells [105], and importantly can become integrated if the infected cell becomes activated before the pre-initiation complex has decayed [38].

## 5. Insertion Site

As a retrovirus, one of the defining features of HIV infection is reverse transcription post-entry into the host cell. Reverse transcription is the process of converting the viral single-stranded RNA genome into a double-stranded DNA genome via the virally encoded reverse transcriptase (RT) (reviewed [106]). The dsDNA is then transported into the nucleus where it utilizes a mix of host- and virus-encoded proteins to permanently insert itself into the human chromosome (reviewed [107]).

### 5.1. Integration Site and Genome Capture Techniques

Integration site determination is complicated by the fact that each provirus has a single integration site and often a single integration per cell [108,109]. Sequencing across the integration site from the provirus into the host DNA can be challenging. Techniques must be able to capture an HIV sequence that is likely to be different than other proviral sequences due to deletions or mutations, and the host sequence adjacent to the provirus is unknown. Even the best designed primers may not be able to bind the HIV sequence. This challenge has led to the development of several methods of integration site analysis (see the Techniques Section below). One major caveat to integration site studies is that most of the technologies do not allow the determination of whether the clonal proviral genomes are replication-competent.

### 5.2. Integration Site Preferences

Retroviruses exhibit some differences in their integration site preferences [110,111]. HIV preferentially integrates into introns of actively expressed genes (Figure 4d,e) [38,65,112,113,114,115]. Unlike many other retroviruses, HIV is capable of infecting non-dividing cells, and the cDNA genome is able to traverse the nuclear membrane through interaction between viral and host proteins [64,116,117,118,119,120]. The nuclear pore complex (NPC) and associated nucleoporin proteins (Nups) aid in facilitating HIV integration into euchromatic regions of the host genome through interactions between the viral capsid and DNA genome-protein integration complex [117]. This results in a strong preference for integration into chromatic regions near the nuclear pores. Integrations near the nuclear envelope is common to lentiviruses with the exception of the gammaretrovirus, M-MLV (Moloney murine leukemia virus) [121].

Mapping the integration site of HIV proviral genomes is exceptionally useful for determining the extent of proliferation each clone has undergone, when clones originated, and the balance between clonal to non-clonal proviruses. This is possible based on the assumption that no two integration events are likely to be exactly the same by chance. With this understanding, one group followed three patients on ART over a period of >10 years, sampling each participant at three timepoints: 1–2 years, 4–8 years, and 11–13 years post ART initiation. All of the participants had identical viral sequences integrated in multiple cells that persisted over all the sampling timepoints, indicating the infected cells underwent proliferation [76]. Similarly, Maldarelli et al. mapped the integration site of HIV proviruses from thousands of PBMCs taken from infected individuals on ART. Approximately 40% of the integrations were in clonally expanded cells. In one of their donors, they astonishingly found as many as half of all integrations sequenced mapped to a single clone [122]. Some groups have also claimed HIV has a strong preference for integration into repetitive DNA, Alu rich, regions of the genome independent of the transcriptional level of the nearby genes, though these findings are controversial [7,112] and others have demonstrated there is no integration preference for many types of repetitive DNA sequences [115].

Many HIV genomes from patients on ART have been found integrated into genes implicated in human cancer, most frequently *BACH2, MDC1* [76,114,115], *MKL2* [7,122], and in genes associated with cell growth and mitosis, including *STAT5B*, *PARP8*, *DDX6*, *PKP4,* and *MAP4* [7,75,76,114,115]. One study found all of the integrations in *BACH2* and *MKL2* were in the sense orientation [115,122] and mapped to introns upstream of the transcription start site (TSS) [122]. When they compared integration sites from acute HIV infection in vitro in both HeLa and CD34+ hematopoietic stem cells, integrations in *BACH2* and *MKL2* were mapped in both the sense and antisense orientation and integrations were dispersed throughout the genes [122]. This suggests that in some cases, there is selective pressure and survival advantage for integrations in a specific orientation and location in genes involved in cell growth and survival even if pre-ART integrations were more stochastic. However, when looking at the overall pattern of integration orientation across all patient data, there seems to be a preference for integration antisense to transcription (Figure 4e) [7,112,115,122,123], suggesting there is much still to be learned about integration site and orientation direction of proviruses and whether proviral integration into genes regulating cell growth affects cell survival. Interestingly, the one integration site from a treated patient sample that has been linked to both an infected CD34+ progenitor cell with naturally high proliferative potential and clonally expanded differentiated progeny was more than 50 KB from any genes (Figure 4c) [11]. Amplification of this provirus by cellular proliferation was thus unlikely to be due to insertion into genes that regulate growth as the insertion site was intergenic and far from a gene. Any cellular proliferation was therefore likely to be from the innate proliferative capacity of the infected cell(s) and not from dysregulation of nearby genes.

Whether HIV actually gains a proliferative advantage from integration into cancer-associated genes is debated [7,115]. Cohn et al. claim the enrichment of clonal integrations in cancer-associated genes is as frequent as integrations in highly expressed genes and thus there is no preferential integration advantage for cancer-related genes. Instead, integrations in cancer-associated genes, transcriptionally active, and genic-regions of the genome decrease in ART-treated compared to untreated individuals [7]. Assuming a preference for integration into cancer and cell growth associated genes, non-AIDS related cancers are surprisingly rare in HIV infected individuals thus far in the epidemic. Cancers associated with HIV infection instead commonly arise from other viral infections such as Epstein–Barr virus [124,125] with rare exceptions [126]. However, pharmacological inhibition of a cancer-related gene decreased HIV-infected cell numbers in vitro [127]. Inhibition of BIRC5 was effective at clearing many HIV infected cells including those latently- and actively-infected. *BIRC5* (Baculoviral IAP repeat containing 5) is a member of the inhibitor of apoptosis (IAP) gene family. Whether the cells exclusively affected by the treatment contain proviral insertions in *BIRC5* or if treatment can broadly clear HIV infected cells remains to be determined. 

### 5.3. Informing Cure Strategies from Integration Sites

Once HIV has integrated into a host chromosome, it persists for the life of the cell. If the HIV genome is replication-competent and if the infected cell is long-lived, there is potential for the provirus to become activated and contribute to viremia at any point. Resting CD4+ T cells have a long half-life [3] and HIV DNA levels remain stable in patients on ART after many years. Resting CD4+ T cells can maintain themselves under a quiescent state or through homeostatic proliferation without reactivation of the latent virus. Hence, any prolonged interruption of therapy creates potential for viral rebound.

Latency reversal strategies have mostly focused on “shock and kill”, an approach broadly described as shocking latent HIV in quiescent cells to transcriptionally awaken. Transcriptionally active HIV produces proteins that are usually either toxic to the cell or reveal the infected cell to the immune system for clearance. The epigenetic state at the site of viral integration is of great interest for treatments involving histone deacetylase inhibitors (HDI) as they have been proposed as a potential therapeutic option for latency reversal (Figure 4a,b). The pan-HDI, vorinostat, has had mixed success in clinical trials for decreasing the latent T cell reservoir [128]. Recent work testing the effectiveness of vorinostat and other pan-HDI on reversing the latent reservoir in HSPCs has demonstrated limited efficacy. Instead, a combination of class-I selective HDI treatments appear to be more effective [129]. Understanding the contribution of integration site selection initially by HIV and over time in vivo has profound implications for curative strategies. As such, there is a database created to collect known integration site information [130].

## 6. Techniques

The phenomenon of clonality can be determined in many ways. The most assured way is to capture the integration site, however, by doing so, information is not captured about the provirus sequence. This is because current laboratory techniques are either optimized to examine the provirus-host chromosome junction (Table 1) or to capture the proviral sequence (Table 2), but not both. Some groups have combined and modified existing techniques to determine the HIV integration site and genome sequence integrity from a single cell (or genomic equivalent) containing an HIV provirus (Table 3) [11,115,131]. Techniques used to measure the latent reservoir are discussed in Table 4. An important caveat to all of the work presented in this review is the method of determining clonality differs between studies and between groups. Sequencing a selection of genes or subgenomic regions to show replication-competency may lead to overestimates of the intact, clonal population as there could be mutations or deletions in other parts of the genome not captured. Only if the integration site as well as the entire HIV genome is sequenced can clonality be definitely determined [9]. Table 1, Table 2, Table 3 and Table 4 contain brief descriptions of a selection of techniques used in some of the studies reviewed here as well as some generally used in the field to study HIV latency and proliferation.

## 7. Conclusions

HIV targets cells based on cell surface expression of CD4 and one of two co-receptors. Many of the infected cells die from infection, but a small fraction of these cells survives and become quiescent, forming the latent viral reservoir. The reservoir is primarily memory CD4+ T cells and hematopoietic stem and progenitor cells. Recently, proliferation of latently-infected cells was identified as a means of maintaining and expanding the viral reservoir through monitoring the unique integration site of each provirus or sequencing full and partial genomes and inferring clonality. Both CD4+ T cells and HSPCs are long-lived and have the potential to proliferate and differentiate. For example, differentiation of an infected HSPC will ensure all daughter cells carry the same provirus in the same genomic location with them, expanding the viral reservoir. Similarly, an infected T cell that proliferates will also carry the provirus into every daughter cell. Some latent clones can expand to significant numbers. The large majority of proliferating latently-infected cells contain a defective provirus; however, a significant minority of clones contain replication-competent provirus. Clonally expanded cells are capable of reactivating and spreading virus, posing a major obstacle to curing HIV. 

The impact of proliferation on the viral reservoir is a new and exciting area of research, but there are many outstanding questions to be addressed. New T cell subsets are now implicated in maintenance of the reservoir, particularly T_N_ and T_SCM_. More studies are needed to confirm the relative impact of these T cell subsets as well as the proliferative capacity of T_N_ and T_SCM_ cells. Similarly, several studies have suggested an important role for latently infected HSPCs, but questions remain regarding the extent of their contribution to the HIV reservoir.

Future studies focused on better understanding the role of integration site selection and its effect on the selective amplification of some HIV proviruses are needed to assess the contribution of host genes to the maintenance of the HIV reservoir. Thus far, studies have provided complex and sometimes contradictory evidence about the relative importance of integration site selection on reservoir maintenance.

Given the potential impact of HIV integration on the human genome, the fact that cancer has not yet been associated with HIV infection is remarkable. However, ART has only been available for at most 40 years and it may be too early to tell whether cancers related to HIV will emerge in the future.

Sex-biases in our understanding of the HIV reservoir must also be addressed. Many studies conducted in the developed world are heavily biased towards male participants, which is concerning given the large body of evidence that suggests there are known sex-specific differences for disease progression and response to therapies (reviewed [80]). 

Finally, although much is already known about clonality, integration site, and replication competency in maintaining the viral reservoir (as discussed in this review) there is still much to be discovered. Given the amount of heterogeneity and the low frequency of intact proviral genomes, high-throughput methods are needed to achieve the goals outlined in this review so that future treatment strategies can be effective at eliminating all viral reservoirs.

## Figures and Tables

**Figure 1 viruses-12-00127-f001:**
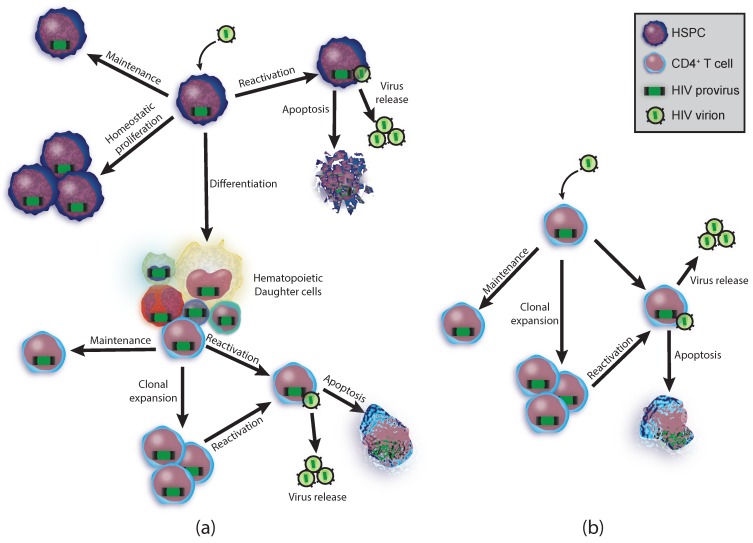
Cellular potential and clonal expansion. (**a**) A hematopoietic stem cell becomes infected with human immunodeficiency virus (HIV) and can follow many fates including maintenance of the population, differentiation, and reactivation (top). Some of the daughter cells can clonally expand (bottom); (**b**) CD4+ T cell fates. A CD4+ T cell becomes infected and can maintain itself, clonally expand, and reactivate to produce more viruses.

**Figure 2 viruses-12-00127-f002:**
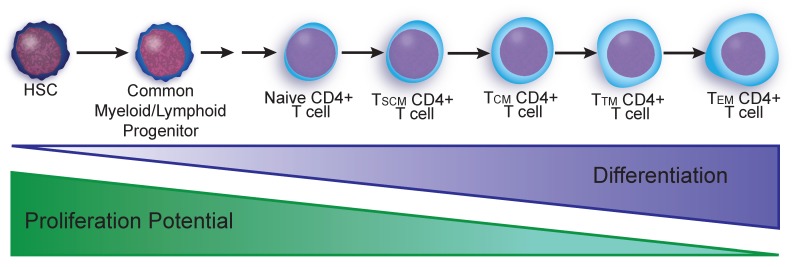
Differentiation and proliferative potential of the latent reservoir. Hematopoietic stem cells (HSC) are multipotent and differentiate into both common myeloid and lymphoid progenitors. Through several more steps of differentiation, naïve CD4+ T cells emerge from the lymphoid lineage as the least differentiated of the terminally differentiated T cell subsets. T cells undergo successive rounds of activation and differentiation through memory stem cells (T_SCM_), central memory (T_CM_), transitional memory (T_TM_), to effector memory (T_EM_).

**Figure 3 viruses-12-00127-f003:**
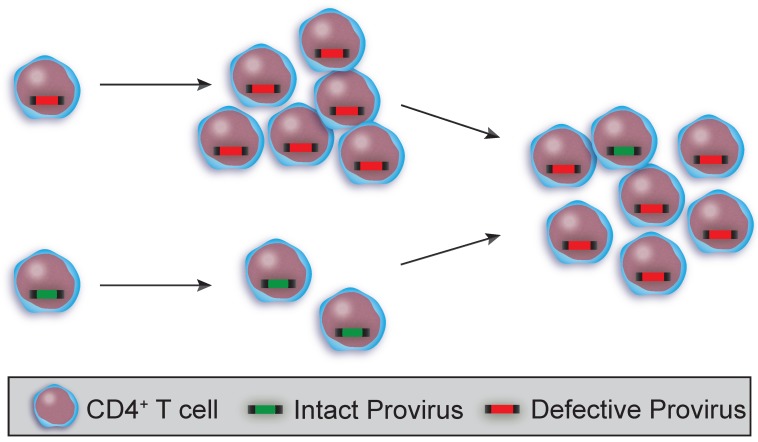
Proliferation of intact and defective proviruses build and maintain the viral reservoir. Cells with intact proviruses (green bar) can sometimes proliferate and expand but do not survive as abundantly as cells harboring defective provirus (red bar) and eventually decrease over time. Proliferation is more abundant for cells with defective proviruses.

**Figure 4 viruses-12-00127-f004:**
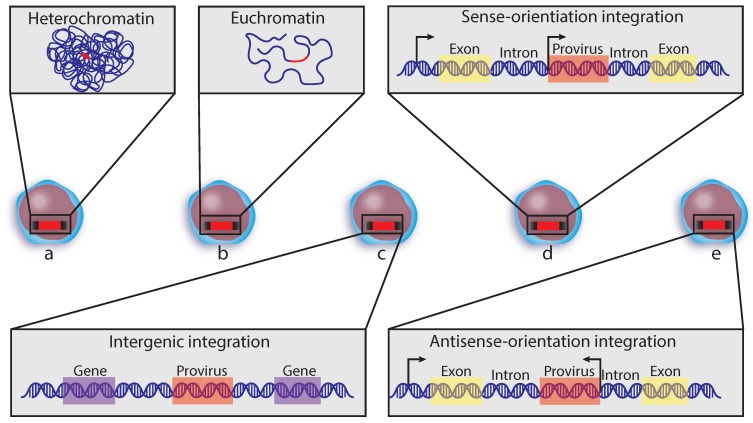
Transcriptional orientation and chromatin states of proviruses. HIV proviruses (red bars) can be found throughout the genome, in both inactive regions of the host genome like heterochromatin (**a**) or actively expressed regions such as euchromatic regions (**b**). They can also be detected in intergenic regions (**c**) but are primarily found as integrations within introns of actively expressed genes (**d**,**e**) and can integrate in either the sense- or antisense-orientation regardless of their genomic location (**d**,**e**). Arrows indicate TSS and direction of transcription.

**Figure 5 viruses-12-00127-f005:**
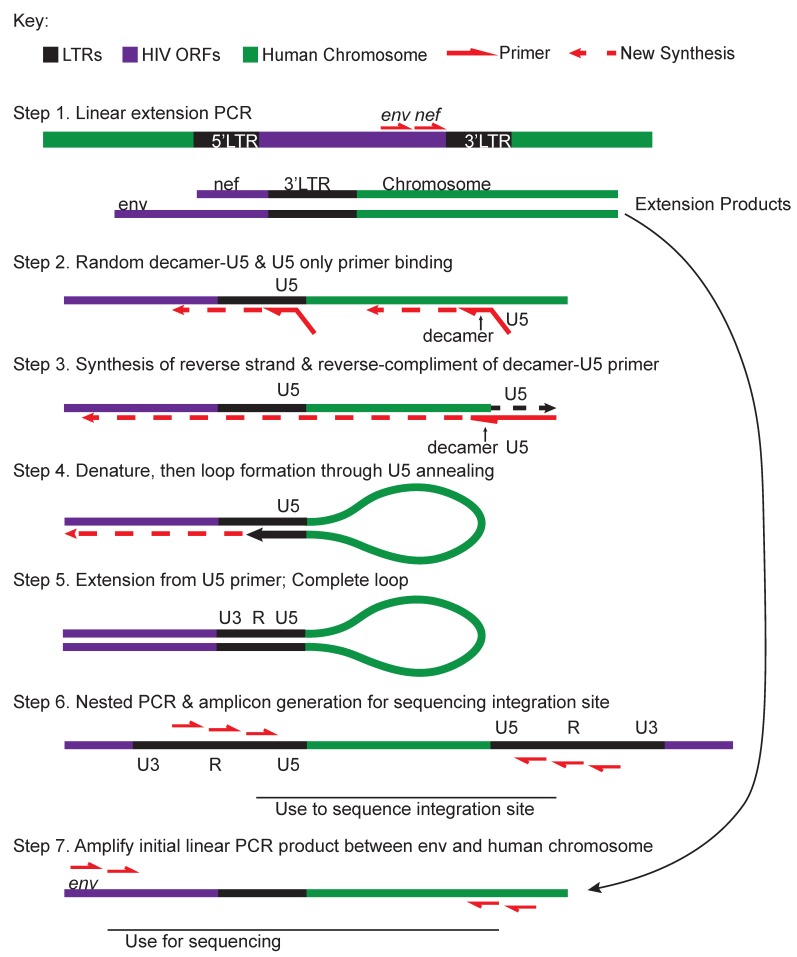
Integration site loop amplification (ISLA). Modified from [76]. Black segments represent long-terminal repeat (LTR) regions; purple, HIV open reading frames (ORFs); green, human chromosomal DNA.

**Table 1 viruses-12-00127-t001:** Integration Site Techniques.

Technique	Description	Advantages	Limitations
**Modified TC-Seq** [7,122]	Sonicate genomic DNA into small fragments. End-repair DNA fragments. Add dA-linker to 3’ ends. Attach linkers to 3’ ends using dT. Perform Nested PCR on fragments. Attach Illumina sequencing adaptors. Paired-end Illumina sequencing across integration sites.	High-throughput integration site analysis Clonality determined through matched integration sites	Provirus sequence cannot be determined
**ISLA (Integration site loop amplification) (Figure 5)** [76]	Linear extension PCR products from the 3’end of HIV into chromosome are amplified with a random decamer complementary to the host genome with a U5-priming sequence-tail. The U5 primer is used as a reverse primer to convert the ssPCR into a dsPCR of 3’HIV-chromosome product. Linearized products produce a genetic barbell with LTR sequence on either side of human sequence which can be PCR amplified and prepared for NextGen sequencing.	High-throughput integration site analysis Clever usage of random decamer-U5 primer for sequence generation across integration sites	Provirus sequence integrity cannot be determined More complicated method compared to others Requires multiple PCR steps
**Circular Template** [11,132]	PCR amplification and sequencing across the insertion site starting from genomic DNA from HIV infected cells. A restriction enzyme (PstI) creates small provirus-host DNA fragments that can be ligated together into circular units and amplified using HIV-specific primers oriented in opposing directions across the integration site, then sequenced by Sanger sequencing.	Very accurate integration site determination Primers for amplification designed for highly conserved genomic regions	Very low-throughput detection of integration site Primers may not recognize all proviruses Requires multiple PCR steps

**Table 2 viruses-12-00127-t002:** Genome Amplification and Sequencing.

Technique	Description	Advantages	Limitations
**FLIPS (Full-length individual proviral sequencing)** [15]	An assay used to capture nearly full-length genomes by two non-overlapping, nested-PCR reactions amplifying from the U5 region in the 5’ and 3’ LTRs. PCR products are sequenced by paired-end Illumina sequencing using the Nextera library preparation kit (Illumina). Reads are mapped by de novo genome assembly.	Captures nearly full-length genomes Can be used to determine replication-competency Can be used to determine clonality	Does not capture all of the LTR sequences
**SCS Assay (single-cell sequencing)** [108]	CD4+ T cells are FACS sorted from HIV-infected patients into a 96-well plate such that there is no more than one cell in any one well. Once the cell in each well is lysed, the DNA is distributed from one well into 10. PCR is used to amplify HIV DNA between *gag-pol*, identified through gel-electrophoresis, and sequenced. Sequenced products can then be aligned to a reference HIV genome.	Primers target highly conserved regions and amplify highly variable regions to identify similarities between HIV sequences Captures all proviral sequences in a cell	Clonal sequences will be overestimated by sequencing partial genomes Deletions and mutations could be found outside of the amplified region Does not capture entire genomes
**SPS/SGS (Single-genome (provirus) sequencing and single genome amplification)** [67,133,134].	Virions collected from the peripheral blood or cultured media containing HIV virion-producing cells are collected, lysed, and converted to cDNA. The cDNA is serially diluted and used for qPCR with primer probe directed at *pol* (SGS) or *env* (SGA). Once HIV-containing samples are identified, the cDNA is then subjected to PCR amplification between *gag-pol* or *env* and sequenced.	Some versions of the protocol capture nearly full-length genomes (nine ORFs) Can be used to determine replication-competency Can be used to determine clonality	Not all versions of the protocol sequence compete genomes Whole HIV genomes assembled from multiple PCR fragments Sequencing requires active HIV expression

**Table 3 viruses-12-00127-t003:** Combined genome amplification and integration site analysis.

Technique	Description	Advantages	Limitations
**Matched integration site and proviral sequencing (MIP-Seq)** [115]	Genomic DNA is isolated from CD4+ T cells, quantified using ddPCR for viral *gag*, and diluted to single proviral genomes based on ddPCR and Poisson distribution. This is followed by multiple displacement amplification (MDA) and whole genome amplification (WGA), generating 1,000-10,000 copies of gDNA. Amplified gDNA is divided such that some is used for NFL sequencing [14] and some for integration site analysis (ISLA; Table 1; Figure 5) or other methods.	Allows for the determination of both the provirus sequence and the integration site from the same cell Integration site analysis is flexible depending on the user-preferred technique Combines techniques already published in the literature for both integration site and genome sequencing	Involves many steps in order to determine the integration site and genome sequence Relies on ddPCR for initial quantification of DNA Must copy the entire human genome, which can lead to errors Sequencing techniques introduce errors that can be mistaken for real mutations
**Multiple-displacement amplification with single-genome sequencing (MDA-SGS)** [131]	Genomic DNA is extracted from PBMCs or other primary cells and diluted across a 96-well plate. Whole-gDNA is amplified in-well using MDA. MDA wells are screened for proviruses of interest using SGS (subgenomic fragments) from P6 through part of RT. Then integration sites are determined using modified TC-Seq (Table 1) followed by NFL amplification using Sanger sequencing or PacBio sequencing.	Allows for the determination of both the provirus sequence and the integration site from the same cell Combines techniques already published in the literature	Involves many steps in order to determine the integration site and genome sequence Must copy the entire human genome, which can lead to errors Sequencing techniques introduce errors that can be mistaken for real mutations Custom pipeline needed to determine intactness of HIV genome

**Table 4 viruses-12-00127-t004:** Measuring the Latent Reservoir.

Technique	Description	Advantages	Limitations
**ddPCR (droplet digital PCR)** (**reviewed** [135])	Target molecules are emulsified into thousands of nanoliter droplets and amplified by PCR using a primer-probe set. Droplets containing HIV genomic material that fluoresces above a certain threshold will be considered positive. The ratio between the positive and negative droplets is used to calculate the absolute number of starting molecules using a Poisson distribution.	Very accurate and versatile Can be used to measure both viral RNA and DNA Does not need a standard curve Produces absolute quantification More reproducible than standard qPCR	Technically difficult and thresholds are set by the user High false-positive rate
**IPDA (intact proviral DNA assay)** [136]	Uses two amplicons covering the packaging signal (ψ) and *env* region and ddPCR to designate deleted proviruses as defective. In parallel, multiplex PCR is performed with two unique primer-probe sets targeting ψ and *env* with unique labeling probes and measured by ddPCR. Primer-probe sets amplify validated, highly conserved regions of the genome. Droplets are scored based on expression of combinations of probe fluorescent patterns.	Primers are pre-designed to recognize most HIV sequences Can quickly determine if a provirus or RNA genome is likely to be intact or defective Primer-probe sets can determine the level of intactness or defectiveness of most proviral and RNA genomes	The entire viral genome is not sequenced and can therefore mis-classify HIV genomes as intact Over-estimates the number of intact viral genomes Requires ddPCR
**TILDA****(Tat/Rev induced limiting dilution assay)** [137]	CD4+ T cells are stimulated in vitro (PMA and ionomycin) to maximally produce *tat/rev* transcripts. Cells are serially diluted as replicates. Real-time PCR with primer-probe pairs are used to quantify inducible viral RNA. The frequency of cells with inducible HIV RNA can then be calculated from the number of positive wells at each dilution by the maximum likelihood method.	Measures latent reservoir from total CD4+ T cells Viral transcripts detected without RNA extraction Requires small blood sample Less labor and time required compared to similar protocols	Only measures genomes capable of being reactivated Not all latent infections can be reactivated and measured [65] Assumes multiply spliced genomes indicate replication-competence
**QVOA (quantitative viral outgrowth assay)** [138,139]	Culture method to quantify the replication-competent viral reservoir. HIV(+) donor rCD4+ T cells are cultured with irradiated PBMCs and CD4+ T cells from an HIV(-) donor and stimulated (PHA; IL-2). Replication -competent virus can spread to HIV(-) CD4+ T cells, amplifying the infection, allowing detection and quantification of viral outgrowth.	Detects and quantifies replication-competent virus Measures both pre- and post-integration latency Well-established and widely used protocol	Only measures genomes capable of being reactivated Not all latent infections can be reactivated and measured Underestimates the size of the latent reservoir Requires allogenic donor lymphoblasts for spreading infection
**mQVOA (Modified quantitative viral outgrowth assay)** [140]	A more sensitive adaptation of the gold-standard assay; CD4+ T cells from patients are serially diluted and stimulated (αCD3/CD28 antibodies). MOLT-4/CCR5 cells are co-cultured with the primary cells. HIV RNA is extracted, and RT-qPCR is performed to amplify *pol*. The number of wells positive for HIV RNA at each dilution level are used to determine the infection frequency by maximum likelihood estimate.	Does not require allogenic donor lymphoblasts for spreading infection qPCR used for quantitative measurement of the viral RNA products	Only measures genomes capable of being reactivated Not all latent infections can be reactivated and measured Underestimates the size of the latent reservoir
**dQVOA (differentiation culture quantitative viral Outgrowth assay)** [141]	Measures the impact of T_EM_ differentiation on induction and outgrowth of replication-competent HIV. rCD4+ T cells from patients are activated through culture with a differentiation cytokine cocktails to drive cells towards the T_EM_ terminally differentiated subset. Cells are distributed at limiting dilutions and cultured in differentiation cytokines, then activated. Titer measured by p24 ELISA.	Higher viral titer induction rate compared to QVOA Higher frequency of latent cell activation over QVOA Does not require allogenic donor lymphoblasts for spreading infection	Only measures genomes capable of being reactivated Not all latent infections can be reactivated and measured Underestimates the size of the latent reservoir

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
