# Peer review of "The Impact of Cellular Proliferation on the HIV-1 Reservoir"

_viruses, 2020, doi:10.3390/v12020127_

Round 1

Reviewer 1 Report

The review article on proliferation of HIV-1 infected cells in individuals on ART by Virgilio and Collins is very thorough and well written. Without a doubt, the review article will be a significant contribution to the field. There are a few modifications and additions, however, that should be addressed prior to publication. 

The statements about hotspots for integration into Alu regions and cancer-related genes are not correct. The one study that claimed preference for integration into Alu regions was due to PCR recombination as evidence by the sites having identical sequences at the junction and different sequences thereafter. Analyses of integration sites in the Retrovirus Integration Sites Database (RID) show that HIV does not have preferences for integration anywhere, with the exception of actively expressed genes, including in Alu or in cancer-related genes, but that there is selection for integration events into proto-oncogenes and, possibly, other genes that regulation the cell cycle. A discussion of new methods that can sequence the full-length HIV and their sites of integration is missing (Einkauf, et al. JCI 2019 and Patro, et al. PNAS 2019) and of their findings. There are several comments that identical HIV sequences have identical integration sites. The only study addressing this question is Patro, et al. in which he finds that identical sub-genomic HIV sequences often have different sites of integration. It is clear that more studies are needed to determine if full-length HIV sequences have identical sites of integration or not. It is premature to make statements claiming that they do. There are many statements claiming clonal proliferation based on identical HIV sequences. Integration sites are needed to identify infected cell clones. These statements should be softened. More references are needed throughout including citations of the proliferative potential of the different cell subsets including naïve cells. Some immunologists would argue that naïve cells do not proliferate. More citations should be included for studies on lack of HIV evolution during ART in tissues as well. Line 153-154 claim that LRAs lessen the viral reservoir. I don’t think there is convincing evidence to support this statement. LRAs may induce the proliferation of infected cells and may even increase the reservoir size despite proviral activation. The statement that the integration site of the AMBI-1 provirus is not known is not correct. The integration site is known but it is in a region of the human genome that is mapped to two different places in the reference. Also, the AMBI-1 provirus was not cloned but the PCR product was transfected in bulk to ensure that there were no point mutations. They claim that intact proviruses decline with time on ART. There is only some evidence that this decline MAY occur in men. In women, intact proviruses do not decrease and may even increase with time on ART and with age (Karn, HIV Persistence Workshop 2019). Line 318 implies that the HIV DNA is translocated to the nucleus but it is actually the viral core that is translocated. Line 388-389, it is too early in the HIV epidemic to know if cancers are associated with HIV. If it occurs in <4% in 40 years, for example, patients have not yet been suppressed on ART in sufficient enough numbers yet to observe it. I would soften the statement that HIV is not associated with cancers to say that it has not been observed with HIV.

Minor comments:

There is a typo in line 179 Replication-competent provirus (hyphen is missing throughout) Latently-infected cells (hyphen is missing throughout) Some abbreviations are missing in the abbreviation section (i.e. TH, ART, PV)

Reviewer 2 Report

This manuscript summarizes the state of knowledge on cellular reservoirs of HIV-1, highlighting their proliferative and differentiation capacities. The review includes a brief description and advantages/limitations of techniques that have been used for the characterization of the major obstacle to viral eradication.

The characterization of HIV reservoirs is a current topic, this review will be appealing for a wide audience interested in varied aspects of HIV. It is well structured and easy to read. Authors accurately examine recent studies and detail pros and cons of each approach.

I suggest minor changes to be consider, in order to improve the quality of the article:

In the Abstract section, it should be added a sentence underlining that clonality and proviral integrity could interfere with reliability evaluation of HIV reservoirs with most available techniques. As it stands, this important conclusion of the review is missing here.

Line 53: 2. Cell type …. and growth stimulus of HIV reservoirs

Line 54: 2.1 The HIV infection “in patients”.

Lines 127-131 refer to an extra information that is superfluous in the context of this review.

Line 151: it would be helpful to add that vorinostat is an inhibitor of histone deacetylase

Line 153: Therefore, some cellular stimulation signals- and epigenetic drugs-are capable…

Lines 167-170: reference 21?

Line 179: proliferative

Line 193: Please, since you use it bellow, introduce here the acronym (MSD).

Paragraph 3.1 should be carefully edited, I list some suggestions to consider:

Line 205: Please, add reference (14?)

Line 206: with 11%?

Line 208: The intact, clonal: this comma should be removed.

Line 213: Please, add reference (15?) at the end of this sentence.

Line 220: Please, for this first time define PV.

Line 230-231: Please add the reference (71?)

Lines 424-425: This sentence should be rephrased

Conclusion Section should mention where the further research can be done
